# Modelling motility of *Trypanosoma brucei*

**Florian A. Overberg**[1], **Narges Jamshidi Khameneh**[2], **Timothy Krüger**[2], **Markus Engstler**[2], **Gerhard Gompper**[1], **Dmitry A. Fedosov**[1]*

**1** Theoretical Physics of Living Matter, Institute for Advanced Simulation, Forschungszentrum Jülich, Jülich, Germany, **2** Department of Cell and Developmental Biology, Biocenter, Julius-Maximilians-Universität of Würzburg, Würzburg, Germany

* d.fedosov@fz-juelich.de

**Data availability statement:** All relevant data are within the paper, its Supporting Information files, and on Zenodo at https://doi.org/10.5281/zenodo.14170745.

## Abstract

African trypanosomiasis, or sleeping sickness, is a life-threatening disease caused by the protozoan parasite *Trypanosoma brucei*. The bloodstream form of *T. brucei* has a slender body with a relatively long active flagellum, which makes it an excellent swimmer. We develop a realistic trypanosome model and perform mesoscale hydrodynamic simulations to study the importance of various mechanical characteristics for trypanosome swimming behavior. The membrane of the cell body is represented by an elastic triangulated network, while the attached flagellum consists of four interconnected running-in-parallel filaments with an active travelling bending wave, which permits a good control of the flagellum beating plane. Our simulation results are validated against experimental observations, and highlight the crucial role of body elasticity, non-uniform actuation along the flagellum length, and the orientation of flagellum-beating plane with respect to the body surface for trypanosome locomotion. These results offer a framework for exploring parasite behavior in complex environments.

## Author summary

Trypanosome parasites cause sleeping sickness in humans, and severely affect a few million people worldwide. After infection, parasites are able to propel through and survive in very different environments, in particular blood with its harsh flow conditions. To better understand the mechanisms underlying the motility of this parasite, we establish a numerical model that realistically mimics trypanosome swimming behavior. The parasite model is embedded in a fluid medium and consists of an elastic body and an attached flagellum, whose properties are informed by available experimental measurements. The simulated parasite reproduces well the motion of real trypanosomes, and shows that body elasticity and flagellum beating characteristics strongly influence its motility. This model can readily be employed to study parasite movement in various complex environments, such as different biological tissues.

**Funding:** F.A.O., M.E., G.G., and D.A.F. gratefully acknowledge support by the Deutsche Forschungsgemeinschaft (DFG) within the Priority Programme "Physics of Parasitism" (SPP 2332). N.J.K., M.E., G.G., and D.A.F. acknowledge funding from the ETN "Physics of microbial motility" (PHYMOT) within the European Union's Horizon 2020 research and innovation programme under the Marie Sklodowska-Curie grant agreement No 955910. The funders had no role in study design, data collection and analysis, decision to publish, or preparation of the manuscript.

## 1. Introduction

The protozoan parasite *Trypanosoma brucei* causes African trypanosomiasis, commonly known as sleeping sickness in humans, which affects a few million people in the world [1]. This often lethal illness is a part of a broader group of diseases instigated by the genus *Trypanosoma*, which can infect multiple transmitting organisms and hosts [2,3]. *T. brucei* is transmitted to vertebrates by the tsetse fly, which hosts trypanosomes over a portion of their life cycle [3–5]. The other part of the trypanosome life cycle takes place in animal or human hosts [3,6]. An outstanding ability of these parasites is to survive in very different environments and to negotiate their way toward specific targets or niches [7]. On this way, trypanosomes can adapt their properties (e.g., locomotion, adhesion) to a specific environment and even circumvent the blood-brain barrier, which is not possible for many other pathogens and parasites [4,6,8,9].

Throughout their life cycle, trypanosomes exhibit several distinct morphotypes, which are characterized by different cell-body geometries with an attached flagellum. For example, the bloodstream form of *T. brucei* features a long slender body that becomes thinner towards the anterior end (see Figs 1A and 3). Its flagellum starts from the flagellar pocket near the posterior end of the body, and runs attached along the body length toward the anterior end [10–13]. The flagellum is partially wrapped around the parasite body, and extends beyond the body length with a free flagellum segment at the anterior end of the trypanosome [14]. The flagellum structure consists of a 9 + 2 axoneme of parallelly running microtubules, which are driven by dynein motors, similar to a sperm flagellum [15]. Actuation forces from the molecular motors result in flagellum beating that resembles a snake-like travelling bending wave. A unique feature of trypanosomes is the paraflagellar rod (PFR), which runs parallel to the flagellum within the membrane [5,16]. While the function of PFR remains unclear, it may play a role in guiding the flagellum attachment to the body surface [16–18] or in providing additional stiffness to the flagellum [19].

Snake-like motion of the flagellum enables the locomotion of trypanosomes in fluidic environments. The bloodstream form of *T. brucei* has an average swimming velocity of about 20 $\mu$m/s when swimming persistently in blood [6]. Furthermore, during forward propulsion, the parasite also rotates around its swimming axis, following a helical trajectory. *T. brucei* can also switch its swimming direction backward by reversing the propagation direction of the actuation wave [14]. Due to their microscopic size (about 25 $\mu m$ in length), trypanosomes are classical low-Reynolds-number swimmers, such that inertial effects are negligible [20–22]. Flagellum beating also induces deformations of the cell body, especially to the thinner anterior part. Even though elasticity of the body clearly damps flagellum beating, the role of body deformations for parasite locomotion or any other processes remains unclear.

Trypanosome locomotion enables the parasites to move through and explore different environments. Furthermore, it is suggested that locomotion is necessary for parasite viability [23] and cytokinesis [24], and aids the parasite to evade the host's immune system through a fluid-stress-mediated transport of attached antibodies toward the posterior end, where they are internalized and digested [25]. Clearly, the motion of trypanosomes results from a complex interplay of their multiple mechanical components. While experimental observations provide valuable insights, it is often difficult to manipulate separately different structural components or to isolate various physical effects which contribute to the parasite propulsion. Here, numerical simulations provide a complementary approach which allows for a controlled variation of various mechanical parameters followed by a detailed analysis of their effects on the parasite swimming behavior.

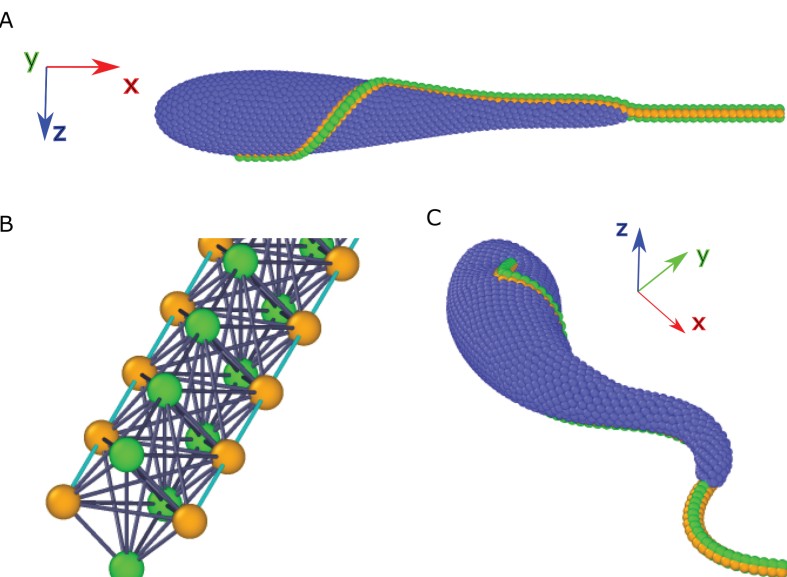

**Fig 1. Sketch of the trypanosome model.** (A) Side view of a trypanosome with an inactive flagellum. Body discretization includes blue particles as well as partially orange particles, which represent the attachment of the flagellum to the body. (B) Flagellum model without a body constructed from four parallel filaments interconnected by springs. Orange particles and cyan springs represent two active filaments (also embedded into the body), which can generate bending deformation, while green particles correspond to the two passive filaments. (C) A swimming trypanosome driven by an active beating of the flagellum.

Several simulation studies [13,14,26] have proposed models of a trypanosome and investigated its swimming behavior. The first detailed model of trypanosome with a spindle-shaped body [26] has predicted a helical trajectory of swimming trypanosome, in agreement with experimental observations [14]. A further study with a better-resolved parasite model [13] has focused on the swimming properties of different trypanosome morphotypes, showing that the bloodstream form of *T. brucei* is a fast swimmer compared to other organisms of similar size. Furthermore, this model has shown that a trypanosome, whose flagellum is partially wrapped around the body as observed in experiments [13], swims faster than a modeled parasite with a unidirectional flagellum configuration. Also, reversal of the swimming direction toward the posterior end has been implemented through the introduction of a counter-propagating actuation wave along the flagellum [13].

In our work, we propose further improvements to the trypanosome model, and study the effects of various mechanical properties of the parasite on its swimming characteristics. The cell body is modeled by a triangulated network of springs, while the attached flagellum is based on a model of four running-in-parallel filaments [27], which allows for a good control of the flagellum beating plane. Our simulations are compared with experimental observations of the bloodstream form of *T. brucei*, yielding a favorable agreement. We demonstrate the importance of body elasticity for trypanosome swimming behavior, which has to be compliant enough to the forces exerted by the flagellum. Furthermore, we explore a non-uniform actuation along the flagellum length, which reproduces well an increase in the beating amplitude toward the anterior end observed in our experiments. Our simulations also show that the beating plane of the flagellum is likely tangential to the body surface, since flagellum

beating in the normal direction poorly captures the rotational dynamics of trypanosomes during swimming. As a result, we clarify the importance of several mechanical characteristics for the swimming behavior of *T. brucei*. Our model can easily be adapted to closely reproduce other trypanosome morphotypes, in order to better understand their swimming behavior. Furthermore, this model can be used to study trypanosome locomotion in more complex environments, such as complex fluids and tissues.

## 2. Materials and methods

### 2.1. Trypanosome model

Our trypanosome model consists of a deformable elongated body and an attached beating flagellum, see Fig 1. Both parasite parts are represented by a collection of interlinked point particles. The modeled parasite is embedded into a fluid, implemented by the smoothed dissipative particle dynamics (SDPD) method [28–30], a particle-based mesoscale hydrodynamics simulation technique. Note that the SDPD fluid fills the whole computational domain, so that it is also present inside the parasite body.

**2.1.1. Trypanosome body.** The cell body of *T. brucei* is modeled by an elastic mesh of point particles, which are homogeneously distributed on a surface of cylindrical symmetry defined by

$$r^2(x) = y^2 + z^2 = c - a(x - x_0)^3 - b(x - x_0)^4,$$ (1)

where $a = 0.18/L_{tryp}$, $b = 0.35/L_{tryp}^2$, $c = 1.96 \times 10^{-4} L_{tryp}^2$, $x_0 = 0.5 L_{tryp}$, and $0 \le x \le L_{body}$, see Fig 1A. Here, $L_{tryp}$ is the total length of trypanosome from the body posterior part to the tip of the flagellum. $L_{tryp} = 24\,\mu m$ for a typical *T. brucei*, and $L_{tryp} = 30$ is selected in simulations, defining a length scale. Eq (1) has been constructed such that the modeled body shape approximates well the main geometrical features of real *T. brucei* cells, whose body has a cylindrical symmetry with a centerline length of $L_{body} = 18\,\mu m$, and the anterior part is thinner than the posterior part. We have selected a maximum radius of $R_{max} = 1.5\,\mu m$, which is similar to that in the previous model of *T. brucei* [13]. From our experimental measurements, the maximum diameter of parasite body generally lies within the range of 2–3 $\mu m$.

The body surface is triangulated by $N_v = 1607$ particles connected by $N_b = 4818$ harmonic springs with a bond potential

$$U_{bond} = \frac{1}{2} k_{s,b} \sum_i^{N_b} (l^i - l_0^i)^2,$$ (2)

where $k_{s,b}$ is the spring stiffness that controls body elasticity, $l^i$ is the spring length, and $l_0^i$ is the individual equilibrium bond length of spring $i$ set after the initial triangulation to impose a stress-free body at rest.

Furthermore, an additional energy potential is introduced to control surface area and volume of the body, [31,32]

$$U_{A,V} = \frac{k_{A,glob}(A - A_0^{tot})^2}{2A_0^{tot}} + \sum_{m \in 1...N_t} \frac{k_{A,loc}(A_m - A_0^m)^2}{2A_0^m} + \frac{k_V(V - V_0^{tot})^2}{2V_0^{tot}},$$ (3)

where $A$ is the instantaneous area of the membrane, $A_0^{tot}$ is the targeted global area, $A_m$ is the area of the $m$-th triangle (or face), $A_0^m$ is the targeted area of the $m$-th triangle, $V$ is the

instantaneous body volume, and $V_0^{tot}$ is the targeted volume. The coefficients $k_{A,glob}$, $k_{A,loc}$, and $k_V$ represent the global area, local area, and volume constraint coefficients, respectively. $N_t$ denotes the number of triangles within the triangulated surface. Note that both the bond potential and the area constraint contribute to area control, since the area-compression modulus is equal to $\sqrt{3}k_{s,b}/2 + k_{A,glob} + k_{A,loc}$. Even though the area constraint might be redundant for large $k_{s,b}$ values, it is important when the body is soft. Furthermore, the local area constraint stabilizes simulations (at least for low values of $k_{s,b}$), since it does not allow very strong local compression or stretching of bonds. The volume constraint is necessary to maintain a desired body volume, since we employ a frictional coupling of the parasite to a background SDPD fluid such that fluid particles can cross the membrane.

To impose bending rigidity of the body, the Helfrich bending energy [33–35] is discretized as

$$U_{bend} = \frac{\kappa}{2} \sum_i^{N_v} \sigma_i (H_i - H_0^i)^2, \tag{4}$$

where $\kappa$ is the bending rigidity of the membrane, $\sigma_i = \sum_{j(i)} \sigma_{ij} r_{ij}/4$ is the area corresponding to vertex $i$ in the membrane triangulation, $H_i = \mathbf{n}_i \cdot \sum_{j(i)} \sigma_{ij} \mathbf{r}_{ij}/(\sigma_i r_{ij})$ is the mean curvature at vertex $i$, and $H_0^i$ is the spontaneous curvature at vertex $i$. Here, $\mathbf{r}_{ij} = \mathbf{r}_i - \mathbf{r}_j$, $r_{ij} = |\mathbf{r}_{ij}|$ and $j(i)$ corresponds to all vertices linked to vertex $i$. $\sigma_{ij} = r_{ij}(\cot\theta_1 + \cot\theta_2)/2$ is the length of the bond in the dual lattice, where $\theta_1$ and $\theta_2$ are the angles at the two vertices opposite to the edge $ij$ in the dihedral.

**2.1.2. Flagellum model.** We adopt a flagellum model, which has been used to simulate sperm flagellum [27,36]. The flagellum is constructed from four filaments which are arranged parallel to each other, see Fig 1B. The equilibrium length of each spring within the filaments of a straight flagellum is $s_0$, which is also the equilibrium length of the diagonals forming a square segment within the cross-section of the flagellum consisting of $N_{seg} = 76$ segments. For structural stability, several diagonal bonds are also introduced, including diagonals within all outer square faces as well as internal diagonals connecting the two passive or active filaments. All bonds within the flagellum structure are implemented through a harmonic potential (see Eq 2) with a spring stiffness $k_{s,f}$, which determines the bending rigidity $K$ of the flagellum (see S1A).

The flagellum is embedded into the body by incorporating two opposing filaments into the body mesh, as shown in Fig 1A. The embedding path of the flagellum along the body is informed by our microscopy observations. The flagellum originates from the flagellar pocket located at the parasite surface 3 $\mu$m from the posterior end measured along the body axis. The first short section runs straight on the body surface, corresponding to 2 $\mu$m length along the body axis. The next flagellum section wraps around the body starting at 5 $\mu$m and finishing at 8.6 $\mu$m along the body axis, completing a half rotation. The final section of the flagellum continues straight on the body surface until the detachment point at 17.8 $\mu$m measured along the body centerline, and finishes with a free straight part of about 6.2 $\mu$m in front of the body's anterior end. The total length of the flagellum is approximately $L_{flag} = 22.4\,\mu m$ and its radius is estimated to be $r_{flag} = s_0/2$.

To generate a bending wave, two opposing filaments in the flagellum structure are made active by prescribing space- and time-dependent changes in the equilibrium lengths $s_0^i(s,t)$ as [27]

$$s_0^i(s,t) = \hat{s}_0^i + a_b \sin\left(2\pi\left(\frac{s}{\lambda_{in}} - ft\right) + \phi_0\right), \tag{5}$$

where $\hat{s}_0^i$ is the initial equilibrium length of spring $i$ along one filament, $a_b$ is the amplitude of the actuation wave, $s$ is the curve-linear coordinate along the filament, $\lambda_{in}$ is the wave length, $f$ is the wave frequency, and $\phi_0$ is a phase shift. One of the active filaments assumes $\phi_0 = 0$, while the other filament has $\phi_0 = \pi$. This difference in phase shift leads to the contraction of one filament and the extension of the other filament or vise versa, generating a bending wave along the flagellum with a maximum curvature $c_{max} = 2a_b/s_0^2$ (see Fig 1C). Note that $\hat{s}_0^i$ is set individually for each spring, because the path of the flagellum has to conform the body, deviating from a straight configuration.

## 2.2. Simulation setup

Simulations are performed in a box with dimensions $L_x = 2L_{tryp}$, $L_y = L_z = 2/3L_{tryp}$ and periodic boundary conditions in all directions. Simulation parameters of the trypanosome are given in Table 1 both in simulation and physical units. Here, $L_{tryp}$ defines a length scale, $1/f$ is the time scale, and $k_BT$ is the energy scale. The membrane shear modulus $\mu_b$ is related to the spring stiffness $k_{s,b}$ as $\mu_b = \sqrt{3}k_{s,b}/4$ [37].

The trypanosome model is embedded into a fluid represented by the SDPD method [28–30], a mesoscale hydrodynamics simulation technique derived through a particle-based Lagrangian discretization of the Navier–Stokes equation (see S2A for details). Fluid viscosity has been set to $\eta = 3.38 \times 10^5 k_BT/(L_{tryp}^3 f)$, which corresponds to $\eta = 5\ mPa \cdot s$ in physical units.

Flagellum actuation can be performed in two different ways: (i) active filaments are those embedded into the body such that the beating plane is tangential to the body surface (see Fig 1C) or (ii) active filaments are the two not embedded filaments such that the beating plane is normal to the body surface. A base trypanosome model (see S1V) is the one with tangential beating of the flagellum, which is similar to the model in Reference [13]. However, for comparison, several simulations using the model with normal beating of the flagellum are also performed. Each simulation is run for about 20 full beats, corresponding to a total time of $20/f$.

**Table 1. Trypanosome parameters in units of the trypanosome length $L_{tryp}$ and the thermal energy $k_BT$ with the corresponding physical values.** $N_{tryp}$ is the number of particles discretizing the trypanosome, $f$ is the beating frequency, $s_0$ is the distance between two cross-sectional segments of the flagellum, $R_{max}$ is the maximum radius of the body, $K$ is the bending rigidity of the flagellum, $\mu_b$ is the shear modulus of the body, $k_{A,glob}$, $k_{A,loc}$, and $k_V$ are the local area, global area, and volume constraint coefficients, and $\kappa$ is the bending rigidity of the body. In simulations, we have selected $L_{tryp} = 30$, $k_BT = 0.1$, and $f = 0.025$.

| Trypanosome parameters | Simulation units | Physical units |
|---|---|---|
| $N_{tryp}$ | 1799 | 1799 |
| $L_{tryp}$ | 30 | $24\ \mu m$ |
| $f$ | 0.025 | $20\ Hz$ |
| $k_BT$ | 0.1 | $4.1 \times 10^{-21}\ J$ |
| $s_0$ | $0.0125L_{tryp}$ | $0.3\ \mu m$ |
| $R_{max}$ | $6.25 \times 10^{-2}L_{tryp}$ | $1.5\ \mu m$ |
| $K$ | $1.48 \times 10^4 k_BTL_{tryp}$ | $1.46\ nN\mu m^2$ |
| $\mu_b$ | $1.56 \times 10^7 k_BT/L_{tryp}^2$ | $110.85\ \mu N/m$ |
| $k_{A,glob}$ | $9 \times 10^6 k_BT/L_{tryp}^2$ | $64 \times 10^{-6}\ N/m$ |
| $k_{A,loc}$ | $9 \times 10^6 k_BT/L_{tryp}^2$ | $64 \times 10^{-6}\ N/m$ |
| $k_V$ | $2.7 \times 10^8 k_BT/L_{tryp}^3$ | $80\ N/m^2$ |
| $\kappa$ | $500k_BT$ | $2.05 \times 10^{-18}\ J$ |

## 2.3. Calculation of swimming characteristics

To characterise parasite motion, its swimming velocity $v$, rotation frequency $\Omega$ around the swimming axis, beating amplitude $B_0$ and wavelength $\lambda_{out}$ are computed from simulation data. The swimming velocity is calculated from the motion of the center of mass (COM) of the posterior part of the body ($3\mu$m in length), because the posterior part is subject to little deformation in comparison to the other parts of the parasite. Each calculation of the COM is an average over 10 time frames separated by 50 time steps. Then, $v$ is computed from fixed-time displacements of the COM as $v = <v_i>$ with $v_i = |\mathbf{r}_{COM}(t_i + t_{beat}) - \mathbf{r}_{COM}(t_i)|/t_{beat}$, where $t_{beat} = 1/f$ is the time of one flagellar beat, $t_i = it_{beat}$, and $i = 1 \ldots M$.

Calculation of the trypanosome rotation frequency requires to determine a swimming axis, for which the gyration tensor $G_{mn} = 1/N \sum_i^N r_m^i r_n^i$ is computed, where $r_m^i$ are coordinates of an $N$-particle system with the origin in its COM, and $m, n \in \{x, y, z\}$. For the calculation of $G_{mn}$, the two thirds of the flagellum length from the anterior end (including only flagellum particles) are used, because this part of the flagellum exhibits beating nearly in a plane. Beating pattern of the one third of the flagellum at the posterior end is quite complex, as it is wrapped around the body. The two eigenvectors of $G_{mn}$ with the largest eigenvalues define the instantaneous beating plane and the eigenvector with the largest eigenvalue provides the swimming axis. Then, the parasite rotation frequency $\Omega$ is computed from the time-dependent angle $\theta$ of the beating plane with respect to the swimming axis using a linear fit to $\theta(t)$.

The two thirds of the flagellum length from the anterior end are also used for the calculation of the beating amplitude $B_0$ and the wavelength $\lambda_{out}$. We find the minimum and maximum values and positions of the flagellum wave with respect to the swimming axis within the beating plane. The minimum and maximum values and positions of the beating flagellum are computed several times over the course of one beat and averaged over the simulation duration. Then, $B_0$ is a half of the distance between the averaged maximum and minimum values, and $\lambda_{out}$ is twice the distance between two consecutive peak positions. Note that the beating pattern of the flagellum may significantly deviate from a sine function, especially for large beating amplitudes. For small beating amplitudes, $\lambda_{in} = \lambda_{out} + \pi^2 B_0^2/\lambda_{out}$. Therefore, $\lambda_{out}$ cannot be larger than $\lambda_{in}$ due to an effective shrinkage of the flagellum along the parasite length. Furthermore, elasticity of the body damps flagella actuation over the part that is attached to the body.

## 2.4. Experimental methods and setup

The bloodstream form (BSF) of *Trypanosoma brucei* (*MiTat 1.6*) was cultivated in HMI-9 medium at 37° C, 5% $CO_2$. The cultured cells were kept in exponential growth phase (i.e., at concentrations below $5 \times 10^5$ cells/ml). For microscopy, 0.4% A4M methylcellulose solution was prepared in HMI-9, resulting in a viscosity of 5 mPa·s, as measured by the supplier. $5\mu$l of cells were placed between a microscope slide and a $25 \times 40$ mm coverslip for imaging. All observations were conducted at room temperature (23° C). Time series for high-resolution and quantitative analysis of flagellum waves were obtained at a frame rate of 100 fps with a pco.edge sCMOS camera, mounted on a Leica DMI6000B equipped with a 63x / NA 1.3 glycerin immersion objective, using differential interference contrast (DIC) microscopy. Forward-persistent swimmers were selected for motility analysis. The amplitude and wavelength at the anterior part of the cell were measured when the wave was in optimal horizontal planar view, as shown in Fig 6A. These measurements were performed using ImageJ.

## 3. Results

### 3.1. Trypanosome swimming properties

Our experimental observations of *T. brucei* swimming in a fluid with a viscosity of $\eta = 5\,mPa \cdot s$ suggest that the beating flagellum contains two full wavelengths (see Fig 3). Flagellum shape over the first half of $L_{flag}$ from the anterior end closely resembles a non-decaying travelling wave, since this part contains the free end and a short portion attached to the body where it is relatively thin. However, flagellum shape over the second half of $L_{flag}$ near the posterior end is much more complex, because the body significantly damps the actuation forces and the flagellum is subject to a non-planar beating due to its wrapping around the body. To better understand how different swimming characteristics (e.g., flagellum wave shape, parasite velocity, and rotational frequency) of a trypanosome are governed by the parameters of flagellum actuation, the amplitude of the actuation wave $a_b$ and its wavelength $\lambda_{in}$ are varied. Note that in experiments we measure $\lambda_{out}$, $B_0$, $v$, and $\Omega$ from microscopy observations, but we have no access to internal actuation characteristics of the flagellum.

Fig 2A shows that the parasite velocity increases with the actuation amplitude $a_b$. The increase in $v$ is particularly pronounced at low actuation amplitudes, and can be attributed to an increase in the flagellum wave amplitude $B_0$ shown in Fig 2C. The dependence of $v$ at low $a_b$ is also consistent with theoretical predictions [21,27,38] of the swimming velocity of a sinusoidally beating filament $v \propto B_0^2 f / \lambda_{out} \propto a_b^2 f \lambda_{in}^3 / s_0^4$. Here, we used the relations $B_0 = 2a_b \left( \lambda_{in} / (2\pi s_0) \right)^2$ and $\lambda_{out} \approx \lambda_{in}$ that are valid for low $B_0$ values. An initial linear increase of $B_0$ with increasing $a_b$ is confirmed in Fig 2C. The swimming velocity $v$ also increases when $\lambda_{in}$ is increased for a fixed $a_b$ due to an increase in $B_0$. Note that the different $B_0$ curves for $L_{flag}/\lambda_{in} \leq 2.15$ appear to be similar to each other, which is partially due to difficulties in the measurement of beating amplitudes, as will be discussed later.

As the actuation amplitude $a_b$ is elevated, the flagellum shape deviates increasingly from a sinusoidal form. In this range of $a_b$, the swimming velocity reaches a maximum, after which it decreases, as shown in Fig 2A for large $\lambda_{in}$ values. Note that the flagellum beating amplitude has a geometrical limit of $B_0 < \lambda_{in}/4$ due to a fixed length of $L_{flag}$, and thus, $B_0$ cannot always increase in response to an increase in $a_b$. Furthermore, the generated propulsion is an integral over the flagellum length, where local contributions depend on the angle between the local tangent of the flagellum and the parasite swimming direction [20,21]. At large beating amplitudes, the local tangent angle is close to $\pi/2$ over a significant portion of the flagellum, which is unfavorable for propulsion.

Fig 2B presents the rotation frequency $\Omega$ of the parasite for various $a_b$ and $\lambda_{in}$ values. Trypanosome rotation becomes faster with increasing the actuation amplitude, which is attributed to an increased deformation of the body by the beating flagellum, as it introduces an increasing helix-like chirality along the parasite length that enhances its rotation. The rotation frequency $\Omega$ depends weakly on $\lambda_{in}$, which is more pronounced at small actuation amplitudes $a_b$. Fig 2D shows that $\lambda_{out}$ decreases with increasing $a_b$. As expected, $\lambda_{out} \approx \lambda_{in}$ at low $a_b$, while $\lambda_{out}$ becomes significantly smaller than $\lambda_{in}$ with increasing actuation amplitude due to the beating-mediated shrinkage of the flagellum along the parasite swimming direction. Note that a few $\lambda_{out}$ values at low $a_b$ are slightly larger than $\lambda_{in}$, because of errors in the analysis associated with difficulties to properly detect the wave peaks at low amplitudes.

The horizontal green lines in Fig 2 mark experimentally measured average values of the corresponding parameters for *T. brucei* in a fluid with $\eta = 5\,mPa \cdot s$. The green shaded areas indicate standard deviations of those measurements. The monitored cells beat on average with a frequency of $f^{exp} = 20.5 \pm 3$ Hz and swim with a velocity of $v^{exp} = 27.6 \pm 5\,\mu m/s$. In simulations with $L_{flag}/\lambda_{in} = 2$, this velocity magnitude can be achieved by setting the actuation amplitude

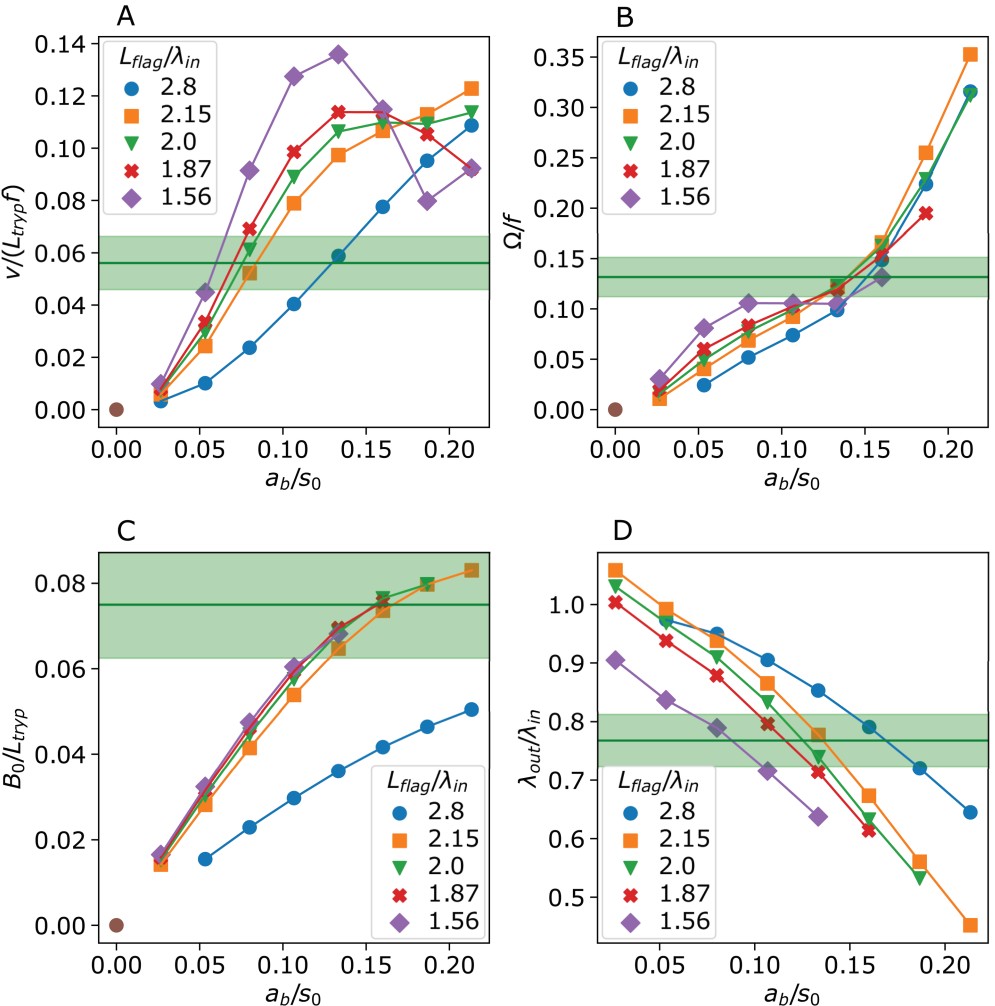

**Fig 2. Swimming characteristics of the trypanosome model as a function of the actuation amplitude $a_b$ and the wavelength $\lambda_{in}$.** (A) Swimming velocity $v$, (B) rotation frequency $\Omega$, (C) flagellum wave amplitude $B_0$, and (D) flagellum wavelength $\lambda_{out}$. Corresponding experimental measurements for *T. brucei* are indicated by horizontal green lines (average values) and shaded areas (standard deviation).

to $a_b/s_0 \approx 0.08$. On the other hand, the experimentally measured rotation frequency $\Omega^{exp} = 2.7 \pm 0.4$ Hz seems to require an almost twice larger actuation amplitude; however, parasite rotation can significantly be enhanced through the relative ratio of the body and flagellum rigidities, as will be shown below. The experimental wave amplitude $B_0^{exp} = 1.8 \pm 0.3 \, \mu m$ is realisable in simulations with $L_{flag}/\lambda_{in} = 2$ for $a_b/s_0 > 0.1$, while the measured wavelength of $\lambda_{out}^{exp} = 8.6 \pm 0.5 \, \mu m$ can be achieved for $0.1 < a_b/s_0 < 0.15$. Fig 3 presents a side-by-side comparison of parasite motion from simulations and experiments over time required for one full rotation (see also S1V and S2V). The correspondence of trypanosome shapes is very good for the parameters $L_{flag}/\lambda_{in} = 2$ and $a_b/s_0 = 0.11$, though the simulated parasite swims slightly faster than that observed experimentally. Thus, if not stated otherwise, further simulations are performed with $a_b/s_0 = 0.11$ to approximate well experimentally observed parasite behavior.

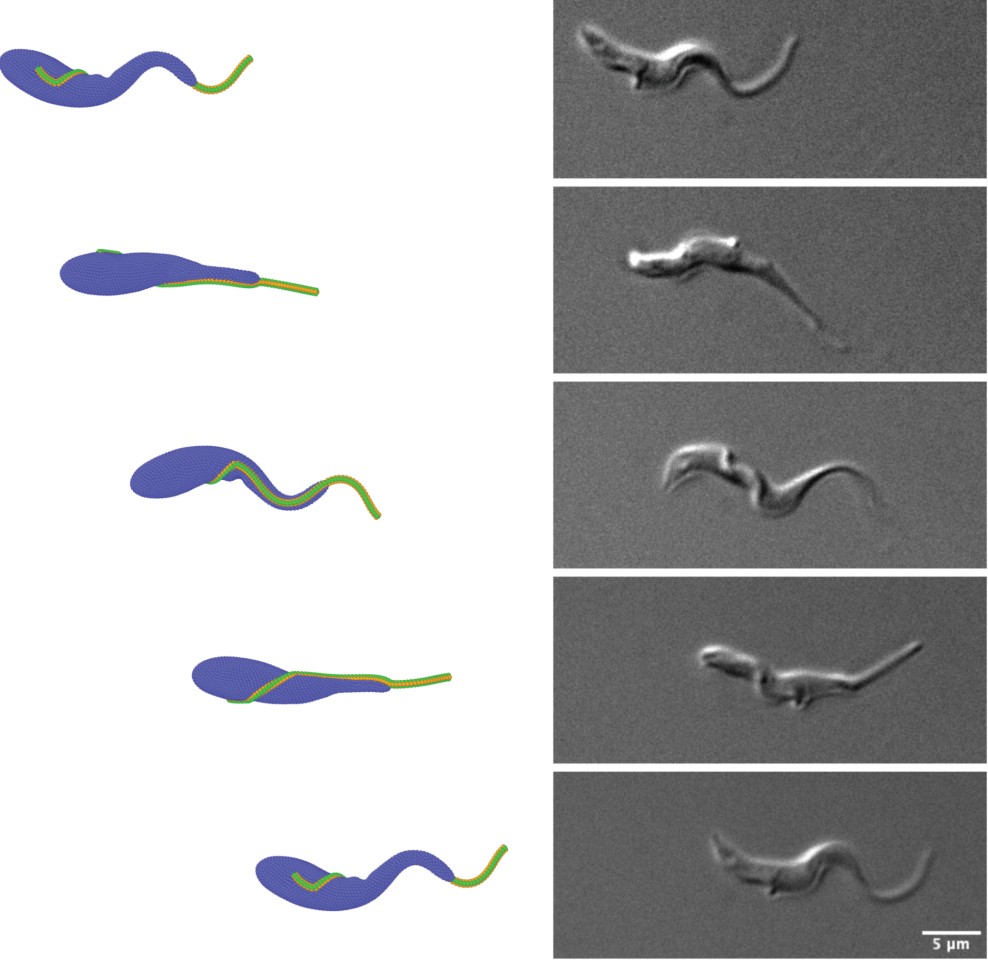

**Fig 3. Side-by-side comparison of** *T.brucei* **swimming from simulations (left) and experiments (right) over the duration of one full rotation around the swimming axis.** In simulations, $L_{flag}/\lambda_{in} = 2$ and $a_b/s_0 = 0.11$ are employed. See also S1V and S2V.

## 3.2. Body and flagellum deformability

The effect of flagellum deformability on its beating shape and dynamics can be captured by the dimensionless sperm number

$$Sp = \left(\frac{\xi_\perp f \lambda_{in}^4}{K}\right)^{1/4},$$

(6)

which represents a ratio of viscous and bending forces, where $\xi_\perp = 4\pi\eta/(\ln(L_{flag}/r_{flag}) -1/2 + \ln(2))$ is the perpendicular friction coefficient per unit length [39]. For a filament with a bending-wave actuation, the swimming velocity is expected to first remain nearly constant at low $Sp$, and then to decay proportionally to $Sp^{-4}$ with increasing $Sp$ [40]. The first regime of a near constant swimming velocity is associated with a quasi-static beating of the filament, for which the bending wave moves instantaneously without significant damping. The regime of

$Sp^{-4}$ is related to an increased viscous damping, which substantially reduces the amplitude of the beating filament [40].

Fig 4A shows the dependence of trypanosome swimming velocity $v$ on $Sp$, where three different parameters $\eta$, $f$, and $K$ are varied. The variation of both $\eta$ and $f$ leads to a persistent decrease in $v$ with increasing $Sp$. For increasing fluid viscosity, there is an increasing viscous dissipation, which reduces the beating amplitude $B_0$ and thus, the swimming velocity. A similar effect takes place when the actuation frequency is increased, as a faster flagellum beating is subject to an increased viscous dissipation that again reduces $B_0$. As expected, the swimming velocity exhibits a plateau at low $Sp$ for varying $\eta$ and $f$, in agreement with theoretical predictions [40] and simulation results [13]. However, the decay in $v$ at larger $Sp$ values is slightly faster than $Sp^{-4}$ when $\eta$ and $f$ are varied, which is likely due to the damping effect by the parasite body. Interestingly, changes in the bending rigidity $K$ of the flagellum (see S3V and S4V) lead to $v \propto Sp^{-4}$ at large enough $Sp$ values, but the swimming velocity does not fully attain a plateau with decreasing $Sp$. Furthermore, $v$ at low $Sp$ when $K$ is varied is slightly larger than $v$ values for the variation of $\eta$ and $f$. Note that the $v$ dependence on $K$ in Fig 4A presents two characteristic slopes. The larger slope at large $Sp$ is attributed to overcoming viscous dissipation with increasing $K$ (or decreasing $Sp$). The smaller slope in $v$ dependence at low $Sp$ is associated with overcoming damping by the body and consequently an enhanced bending of the parasite body as $K$ is increased.

Fig 4B presents trypanosome rotation frequency $\Omega$ as a function of $Sp$ when $\eta$, $f$, and $K$ are varied. Similar to the swimming velocity, $\Omega$ decreases with increasing $Sp$. For instance, when the fluid viscosity is increased, it slows down not only the translational motion of the parasite, but also its rotation. The dependence of $\Omega$ as a function of $Sp$ is very similar for varying $\eta$ and $f$, except for large $Sp$ values. At large beating frequencies, deformation of the parasite body is different from that at low frequencies, which appears to strongly affect its rotation. A decrease in $\Omega$ for varying $K$ is much faster as a function of increasing $Sp$ in comparison to that when $\eta$ and $f$ are altered. Note that parasite rotation is significantly affected by the relative ratio of body and flagellum rigidities, as will be discussed below.

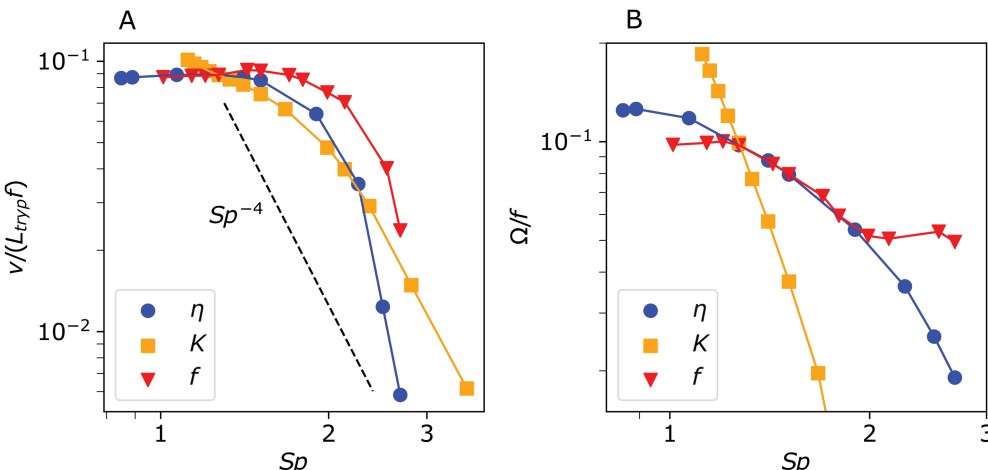

**Fig 4. Parasite swimming properties for different sperm numbers.** (A) Swimming velocity $v$ and (B) parasite rotation frequency $\Omega$ as a function of sperm number $Sp$ obtained from simulations. Three different parameters are varied, including fluid viscosity $\eta$, beating frequency $f$, and bending rigidity $K$ of the flagellum (see S3V and S4V).

As already mentioned, the stiffness of the body also affects trypanosome swimming behavior, whose effect is not included in the definition of $Sp$ in Eq (6). Fig 5 shows various swimming characteristics of a trypanosome as a function of the membrane shear modulus $\mu_b$ (see also S5V and S6V). As the parasite body becomes less deformable (or $\mu_b$ increases), both the swimming velocity $v$ and the rotation frequency $\Omega$ decrease, as shown in Fig 5A. Here, a less deformable body dampens the actuation of the flagellum, which is confirmed in Fig 5B through a reduction in the beating amplitude $B_0$ as $\mu_b$ increases. As a result, both $v$ and $\Omega$ decrease with increasing body rigidity. Interestingly, the rotation frequency is much more sensitive to changes in the body elasticity in comparison to the swimming velocity which decreases only by about 20% for the studied range of $\mu_b$, while $\Omega$ changes almost ten-fold. The moderate decay in $v$ is consistent with a slight decrease in $B_0$, as the body gets stiffer. This is in agreement with the fact that the majority of parasite propulsion is generated by the half of the

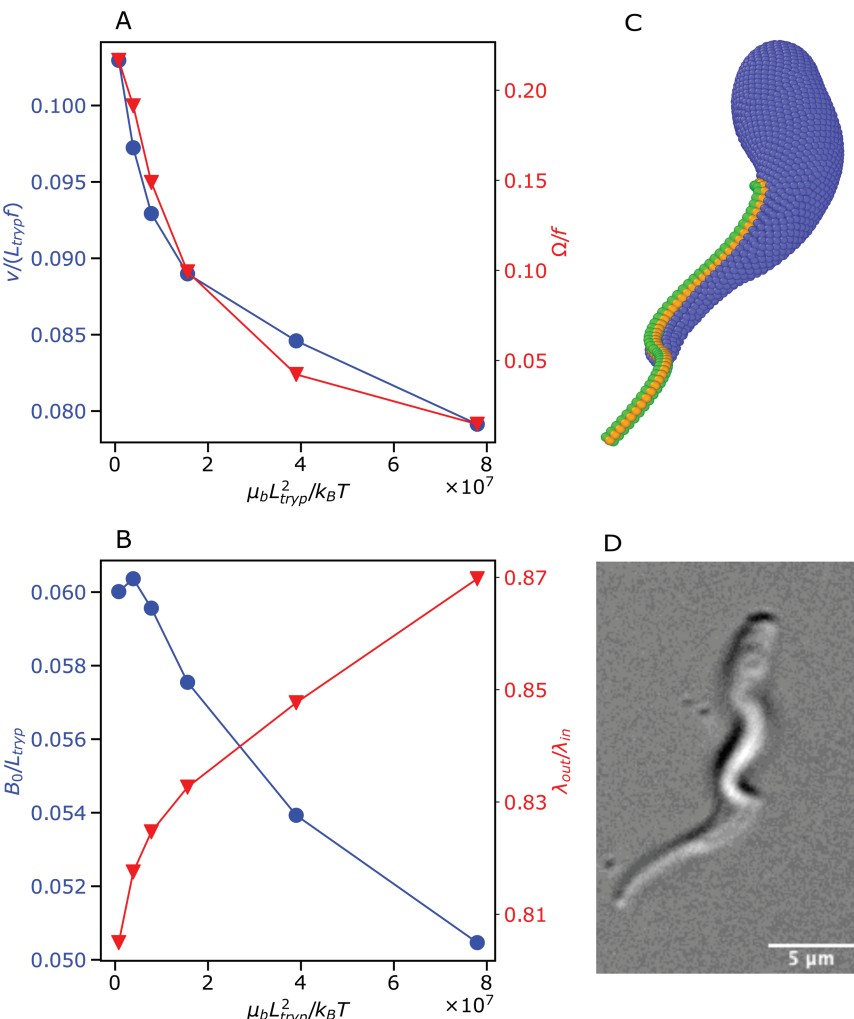

**Fig 5. Parasite swimming characteristics for different body stiffnesses.** (A) Swimming velocity $v$ and rotation frequency $\Omega$ and (B) flagellum wave amplitude $B_0$ and wavelength $\lambda_{out}$ as a function of the body stiffness $\mu_b$. (C) Simulation snapshot of a bent trypanosome with a relatively soft body. (D) Experimental visualization of a banana-like trypanosome shape. See also S5V and S6V.

flagellum length at the anterior end, such that the deformation of the body has a rather minor effect. However, the body deformation plays a substantial role in parasite rotation as it introduces a helix-like chirality into the parasite shape, especially where the body is wrapped by the flagellum. This chirality significantly increases trypanosome rotation frequency, suggesting that parasite rotation is mainly governed by the posterior part of the flagellum and body deformability. Furthermore, a more deformable trypanosome attains a slightly bent overall shape, which is illustrated in Fig 5C. In fact, trypanosome parasites observed experimentally clearly show a banana-like shape during swimming, as shown in Fig 5D. The slight bending of the parasite shape at low $\mu_b$ results from the effective shrinkage of the actuated flagellum, when it is able to deform the body, and leads to a helical trajectory of parasite motion.

### 3.3. Non-uniform flagellum actuation

Experimental observations of a swimming trypanosome show that the wave amplitude at the posterior part is about one-third of $B_0$ at the anterior part, which is illustrated in Fig 6A. This is generally attributed to body elasticity, which may significantly dampen the flagellum wave. However, our simulation results in Sect 3.2 for different body rigidities only partially support this proposition. Even for relatively large stiffnesses of the body, differences in the flagellum

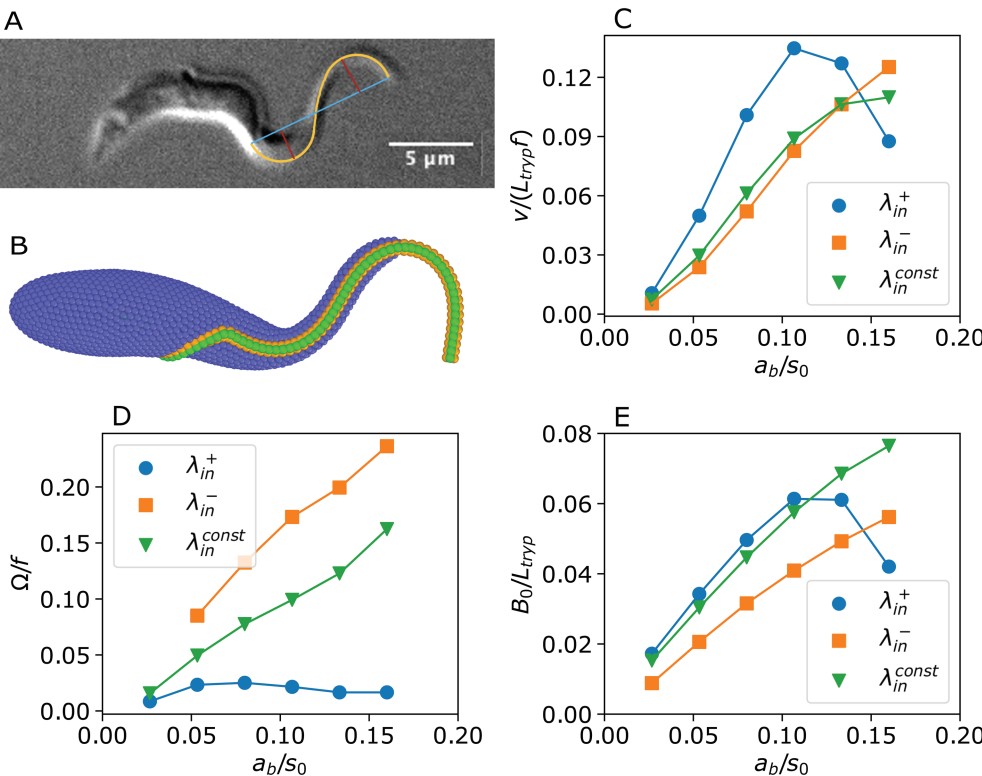

**Fig 6. Swimming properties of a trypanosome with non-uniform flagellum actuation.** (A) Experimental image of a trypanosome which illustrates an increasing wave amplitude toward the anterior end. The red lines mark $B_0$ measurements, while the cyan segment represents the measurement of $\lambda_{out}$. (B) Trypanosome snapshot for a larger $B_0$ at the anterior part in comparison with the posterior part. (C–E) Trypanosome model swimming characteristics for a uniform ($\lambda_{in}^{const}$) and non-uniform ($\lambda_{in,j}^{+}$ and $\lambda_{in,j}^{-}$) flagellum actuation along the flagellum length. (C) Swimming velocity $v$, (D) rotation frequency $\Omega$, and (E) flagellum wave amplitude $B_0$. See also S7V and S8V.

beating amplitudes at the anterior and posterior ends are smaller than those observed experimentally (compare Fig 6A with Fig 1C). It is possible to increase the wave amplitude through the enhancement of the actuation amplitude $a_b$, but this would result in a nearly homogeneous increase of wave amplitude along the flagellum. Another possibility is that the flagellum actuation is not uniform along its length. To enhance the difference in flagellum beating amplitudes between the anterior and posterior ends, a linear change in $\lambda_{in}$ along the flagellum is considered. An increase in $\lambda_{in}$ from the posterior end to the anterior end is implemented as $\lambda_{in,j}^+ = \lambda_{in}^0 + \Delta\lambda_{in}j$, where $\lambda_{in}^0 = 7s_0$, $\Delta\lambda_{in} = 0.092s_0$, and $j$ represents the j-th flagellum segment, counting from the posterior end to the anterior end. A decrease in $\lambda_{in}$ is given by $\lambda_{in,j}^- = \lambda_{in}^0 + \Delta\lambda_{in}(N_{seg} - j)$, which simply reverts the change of $\lambda_{in}$ along the flagellum. The parameters for $\lambda_{in,j}^\pm$ are selected such that the entire flagellum length still contains two waves as assumed before. For the dependence of $\lambda_{in,j}^+$ ($\lambda_{in,j}^-$), the contour length of the first wave at the posterior end is $L_{flag}/3$ ($2L_{flag}/3$), while the contour length of the second wave at the anterior end is $2L_{flag}/3$ ($L_{flag}/3$). Note that the actuation amplitude $a_b$ and the bending rigidity $K$ of the flagellum are kept constant. Fig 6B confirms that the parasite beating shape with $\lambda_{in,j}^+$ agrees well with the experimental image in Fig 6A.

Figs 6C, 6D, and 6E shows various trypanosome swimming characteristics for uniform and non-uniform flagellum actuation as a function of actuation amplitude $a_b$ (see also S7V and S8V). For an increasing wave length $\lambda_{in,j}^+$, trypanosome swims faster than for $\lambda_{in,j}^-$ and $\lambda_{in}^{const}$, because of larger flagellum wave amplitudes $B_0$. Note that for the non-uniform flagellum actuation, $B_0$ is not constant along the flagellum even without the damping effect of the body, so that the values of $B_0$ in Fig 6E correspond to averages, as described in Sect 2.3. However, the rotation frequency $\Omega$ for the case of $\lambda_{in,j}^-$ is significantly larger than that for the case of $\lambda_{in,j}^+$. This is due to differences in body deformation. For the case of $\lambda_{in,j}^+$, the wave at the posterior end is too small to significantly deform the body, resulting is a slow trypanosome rotation. For the case of $\lambda_{in,j}^-$, it is the opposite, and a substantial deformation of the body leads to fast parasite rotation.

## 3.4. Tangential versus normal beating plane of the flagellum

Although the trypanosome model considered so far, with the beating plane tangential to the body surface, already satisfactorily reproduces various parasite characteristics, it is interesting to see how the beating orientation affects trypanosome behavior. For a beating plane perpendicular to the body surface, the two filaments that are not embedded into the body are made active. Fig 7 illustrates two snapshots corresponding to tangential and normal beating with respect to the body surface. For the tangential beating, the active filaments are drawn in orange, while for the normal beating, the active filaments are green (see S1V and S9V).

Fig 8 compares the swimming velocity $v$ and rotation frequency $\Omega$ for the two parasite models with different beating planes of the flagellum. Trypanosome swimming velocity is

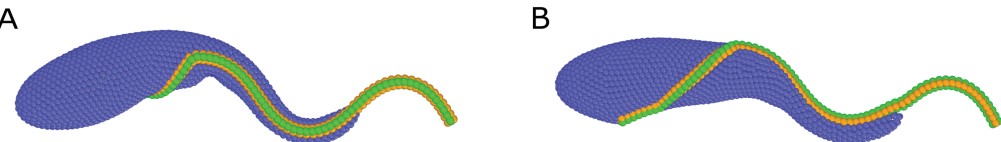

**Fig 7. Tangential versus normal beating plane of the flagellum.** For the tangential beating in (A), the active filaments are drawn in orange, while for the normal beating in (B), the active filaments are green. See S9V.

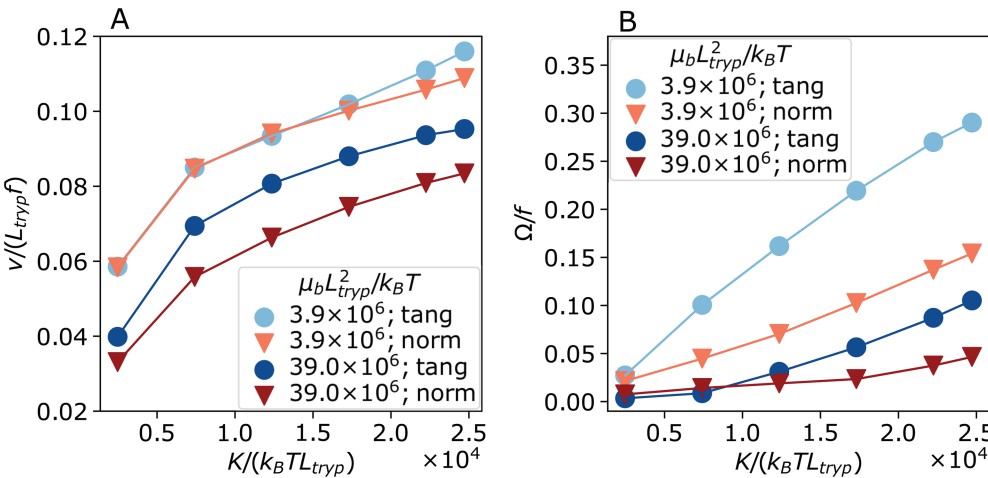

**Fig 8. Trypanosome model swimming characteristics with the flagellum beating tangential (circles) and normal (triangles) to the body surface as a function of bending stiffness $K$ for two body stiffnesses $\mu_b$.** (A) Swimming velocity $v$ and (B) rotation frequency $\Omega$ are shown.

quite similar for the both models as a function of the flagellum bending rigidity $K$. This is due to the fact that the parasite propulsion is primarily generated by the flagellum part at the anterior end, where the effect of the body on flagellum beating is minimal and therefore, the orientation of the beating plane would not be important for propulsion. Fig 8B shows that the rotation frequency is larger for the tangential beating in comparison to the normal beating with respect to the body surface. Our simulations show that the helix-like chirality of the parasite body is stronger for tangential than for normal beating. Furthermore, the model with tangential beating exhibits a slightly bent banana-like shape of the body consistent with the experimental image in Fig 5D, which was discussed in Sect 3.2 and appears when the body is soft enough. Note that the model with normal beating of the flagellum does not exhibit a bent banana-like shape of the body during swimming.

### 3.5. Passive flagellum conformation

The trypanosome model depicted in Fig 1A assumes the flagellum shape wrapped around the body with a straight free end to be its stress-free equilibrium state, which is achieved by setting individually all spring lengths at rest. Even though an unactuated shape of the trypanosome flagellum is not known, it is possible that it has a straight equilibrium shape along the whole length. Fig 9A illustrates a stationary trypanosome shape with a straight equilibrium shape of the flagellum, whose attachment to the body is the same as in Fig 1A. In this case, both the body and the flagellum are subject to non-vanishing elastic stresses, as they balance each other. The main qualitative difference to the case with the wrapped equilibrium shape is that the curved wrapping portion becomes noticeably less bent for the straight equilibrium shape, especially for $\mu_b/K \to 0$.

Fig 9B and 9C compares trypanosome swimming properties for the two equilibrium shapes of the flagellum. Parasite swimming velocity is similar for the both equilibrium shapes of the flagellum, though the parasite is slightly faster for the wrapped equilibrium shape in comparison to the straight flagellum. This is again because the propulsion is primarily generated by the anterior part of the flagellum, where the two equilibrium shapes do not differ much from each other. However, parasite rotation of softer bodies in Fig 9C is much stronger

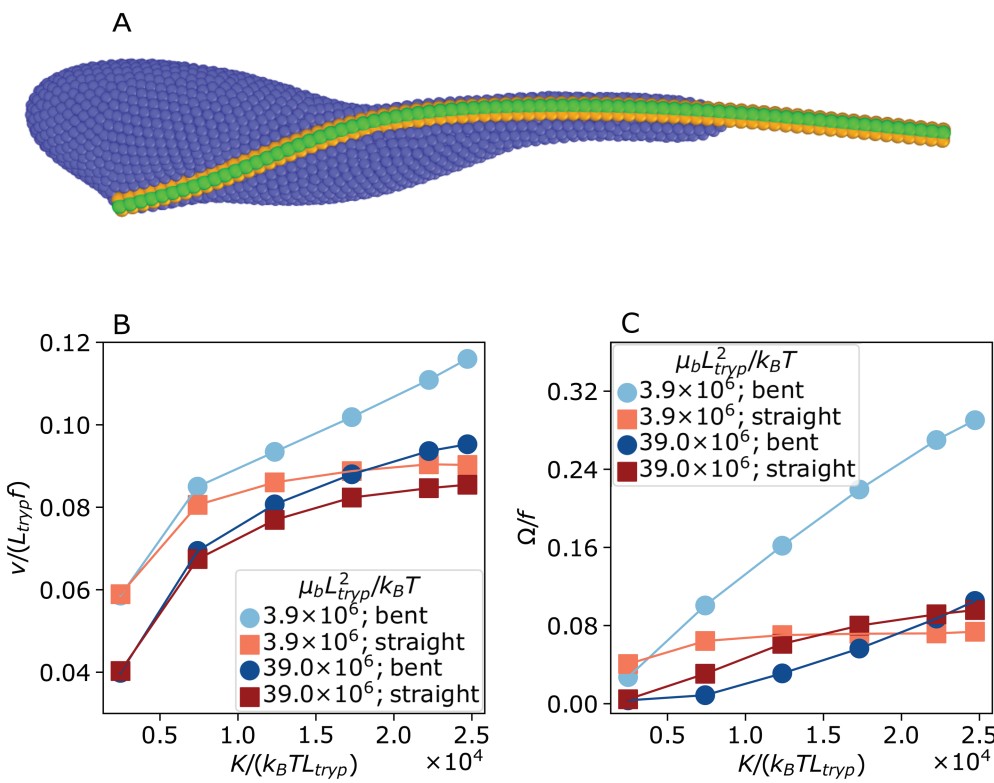

**Fig 9. Effect of passive flagellum conformation on the swimming behavior.** (A) Simulation snapshot of an unactuated parasite with straight equilibrium state of the flagellum. (B–C) Trypanosome model swimming characteristics for a flagellum with bent (circles) and straight (squares) equilibrium states as a function of bending stiffness $K$ for two body stiffnesses $\mu_b$. (B) Swimming velocity $v$ and (C) rotation frequency $\Omega$ are shown.

in the case of the bent flagellum compared to the straight flagellum. For the latter, the rotation is limited, because the flagellum force dominates over the body forces for $\mu_b/K \to 0$ resulting in an effectively straight flagellum. If the body is stiff enough to bend the flagellum, the rotation frequency is comparable and actually partly higher for the straight flagellum. Our simulations also show that the model with the straight equilibrium shape of the flagellum exhibits a banana-like shape to a lesser degree in comparison to the wrapped equilibrium shape.

## 4. Discussion & Conclusions

The described model of *T. brucei* is strongly informed by our microscopy observations, including the shape and size of the parasite, and the path of flagellum attachment along the body. Many other characteristics, such as rigidity of the body and the flagellum, and a force pattern that drives flagellum actuation, are not known. Our trypanosome model is flexible enough to test the importance of several parasite properties for its swimming behavior. Furthermore, it allows us to study theoretical limits of trypanosome propulsion in a Newtonian fluid, and refine physical mechanisms which govern its swimming behavior.

Our simulations support the hypothesis that trypanosome propulsion is primarily generated by the anterior part of the flagellum, while the posterior part that is wrapped around the body has only a minor effect on the swimming velocity *v*. By changing the body stiffness from soft to relatively rigid, we capture a wide range of flagellum-induced body deformations

from strong to weak, which has little effect on $v$. Nevertheless, the relative ratio of flagellum-to-body stiffness has a profound effect on the parasite rotation dynamics. A parasite with a stiff body that is only weakly deformed by the flagellum forces rotates much slower than a trypanosome whose soft body is significantly deformed by the flagellum. Deformation of the body is a primary determinant of the frequency of parasite rotation, as it introduces a helix-like chirality along the body which enhances rotational motion. As a result, the posterior part of the parasite (or more precisely its deformation) governs trypanosome rotation frequency during swimming.

In fact, a soft body is not a necessary condition to have a favorable comparison of rotational frequencies from simulations and experiments. Nevertheless, the forces exerted by the flagellum on the body have to be sufficient to deform it. In our model, flagellum bending rigidity and the strength of actuation are coupled, as both are proportional to the spring constant $k_{s,f}$. Note that it is also possible to change them independently, but it would require the adjustment of $k_{s,f}$ and the actuation amplitude $a_b$ at the same time. Currently, there are no experimental estimates for the body and flagellum rigidities of *T. brucei*. Our choice of the flagellum bending rigidity $K = 1.46\,nN\mu m^2$ (Table 1) is informed by the measurements for echinoderm sperm flagella with $K = 0.3 – 1.5\,nN\mu m^2$ [41]. Since the trypanosome flagellum is accompanied by a paraflagellar rod running in parallel, we chose a $K$ value that represents a relatively stiff flagellum. Note that much smaller values of $K$ are unlikely, because flagellum beating might transit from planar to three dimensional with increasing $Sp \propto 1/K^{1/4}$ due to an elastic instability [42]. However, our experimental observations of trypanosome locomotion indicate a planar beating pattern of the flagellum at the anterior end.

An interesting aspect of trypanosome structure is the equilibrium shape of the flagellum. Does the flagellum conform the body by wrapping around it without significant applied stresses? Or does the flagellum induce torsional and bending stresses on the body, which would be the case of a straight equilibrium shape? Our simulations suggest that the straight equilibrium shape of the flagellum substantially reduces its bent part at the posterior end, where it is wrapped around the body. Even though this does not affect much the swimming velocity, it significantly impairs parasite rotation. Based on these simulation results, of course, we cannot exclude the possible presence of residual stresses in a stationary parasite shape, however, our simulations indicate that the equilibrium shape of the flagellum should be close to the curved shape conforming the body. Note that this would imply a constant bending of the comparably stiff microtubules of the axoneme structure of the flagellum. A possible realization of such a bent shape is that the PFR may have a bent shape, forcing the flagellum to conform the body shape. Even less clear is the actuation pattern of the flagellum or how the actuation forces are distributed along its length. Our basic model prescribes a travelling bending wave along the flagellum, which seems to be a plausible assumption. Experimental images, as in Fig 6A, suggest that the wave amplitude increases as we go from the posterior end of the parasite to its anterior end. The common proposition is that the body strongly damps flagellum beating amplitude at the posterior end. A substantial damping of the flagellum beating is achieved by a stiff trypanosome body in simulations, however, in this case the rotation of the parasite appears to be much slower than that observed experimentally. On the other hand, a soft body that facilitates a parasite rotation frequency consistent with our experimental measurements does not sufficiently damp flagellum beating. One possible proposition is that the actuation along the flagellum length is not uniform and possibly gets stronger toward the anterior end. We have implemented this proposition by making the actuation wave length $\lambda_{in}$ position-dependent along the flagellum length. This modification to flagellum actuation does reproduce well the increase in beating amplitude toward the anterior end. Nevertheless,

it is not possible to confirm or refute this proposition at the current state of knowledge about flagellum actuation.

Furthermore, we have considered different directions of flagellum beating, including tangential and normal orientation with respect to the body surface. Our model allows for an easy bimodal switch of the beating plane, because the flagellum is modeled by four running-in-parallel semi-flexible filaments. As mentioned previously, the swimming velocity is hardly affected by the choice of the beating plane, while the rotation frequency is quite sensitive to this. For the beating actuation normal to the body surface, the parasite swims straight with a relatively low rotation frequency. In contrast, tangential beating with respect to the body surface leads to a helix-like deformation of the body, where it is wrapped by the flagellum, which strongly enhances trypanosome rotation. Furthermore, the tangential beating results in slight parasite bending along its length into a banana-like shape, in agreement with experimental images, as in Fig 5D. The bent parasite shape appears due to the contraction of the actuated flagellum along the trypanosome length in comparison to the straight shape of a non-moving parasite. Furthermore, the banana-like shape results in a helical swimming trajectory of the parasite (i.e., not just a simple rotation around the swimming axis), which is consistent with experimentally observed behavior of trypanosomes [13]. Even though a fixed beating orientation of the flagellum in our model adequately reproduces parasite swimming characteristics, orientation of the beating plane in real trypanosomes is likely more complex [43,44] or might partially be dynamic. While it is known that flagellum bending occurs perpendicular to the midline of the axoneme central pair [45], a detailed information about axoneme orientation for the length of the body is currently not available. Note that our simulation results for two different beating planes already provide some insight into this issue, as we find that the tangential beating deforms the body much less, and therefore also generates less dissipation. We therefore conjecture this to be the preferred orientation if the beating plane were free to adjust itself. Nevertheless, a model with a self-adjusting orientation of the beating plane is worthwhile to implement in the future.

Even though we have considered a number of mechanical characteristics of *T. brucei*, an advantage of our model is that adaptations and further extensions are easily possible. The geometry of the body can be modified by adapting Eq (1). Furthermore, the path of flagellum attachment along the body and the length of free flagellum part at the anterior end can straightforwardly be modified. These modifications should allow simulations of other trypanosome forms, such as a comparatively short and thick stumpy bloodstream form or a slender mesocyclic form in Tsetse fly [4]. For the adaptation of different swimming properties, flagellar beating can be modified by changing the frequency, amplitude, and wavelength of the actuation wave (see Eq 5). A further possible extension of the model is bulk elasticity of the body, which can be implemented as a volume-spanning network of springs. We expect bulk elasticity not to result in a qualitatively different parasite swimming behavior, since it effectively modifies body resistance to deformation similar to our investigation in Sect 3.2. Bulk elasticity has been implemented in the previous trypanosome model [13], though it was primarily used for the stabilization of the body shape. Another important component of the cell structure of *T. brucei* is the sub-pellicular microtubule corset, which runs near the membrane parallel to the body axis [12,46,47]. The presence of microtubules is expected to introduce anisotropic mechanical properties and modify body deformation. Our model already predicts the effect of body deformability on parasite swimming properties, which should be qualitatively similar to that from the microtubule corset. Nevertheless, a model with explicit microtubule corset would be required to quantify its detailed effect on trypanosome swimming characteristics. This can be implemented by introducing semi-flexible filaments attached to the membrane. A further structure that is not explicitly included in our model is the PFR,

which may locally affect trypanosome beating dynamics [16,19,48], since it provides additional bending stiffness. As a result, it may affect the swimming properties quantitatively, but we expect no qualitative changes in the swimming behavior, because the effect of PFR should be qualitatively captured by changing flagellum bending stiffness (see Sect 3.2). Finally, parasite deformation can be affected by viscous dissipation inside the body. In our model, the viscosity inside the body is the same as that of the suspending fluid. A larger viscosity inside would lead to less deformation of the body for the same actuation strength due to larger dissipation. This is similar to a stiffer body (i.e., less deformation), which can significantly reduce parasite rotation, but does not strongly affect the swimming velocity.

Another limitation of performed simulations is that a relatively small simulation domain with periodic boundary conditions has been employed. Thus, the swimming trypanosome interacts hydrodynamically with itself, which may lead to a reduction in its swimming velocity and rotation frequency. We have performed a few simulations with domain dimensions doubled, which have shown a similar swimming behavior of the parasite with a slight increase (less than 10%) in both $v$ and $\Omega$. Note that twice larger domain dimensions in 3D result in an eight-fold increase in computational cost. Furthermore, the analysis of parasite swimming properties based on the gyration tensor is not always reliable, especially in the limits of small and large amplitudes. In those cases, it is difficult to reliably determine the beating plane. However, these limits of flagellum beating represent situations which are generally outside the range of normal trypanosome behavior. Experimentally measured swimming characteristics are of course also subject to errors, which can come due to an imprecision in determination of the beating plane from 2D images.

Existing variations in trypanosome types and species are generally viewed as adaptations to different environments [3,4]. While the developed model can be used to study parasite behavior in various habitats (e.g., blood suspension or tissue), conclusions drawn from the current simulations are limited to the swimming behavior in bulk fluid, and should not be extrapolated to more complex environments. For instance, some species may react to the presence of mammalian cells and alter their motility properties, including adaption of propulsion characteristics as well as changes in the frequency of tumbling events or backward swimming [6,49]. Furthermore, trypanosome parasites may utilize surrounding structures for the enhancement of their propulsion, as suggested in microfluidic experiments with pillar arrays [6,14].

In conclusion, the developed model of trypanosome properly captures various swimming properties of the parasite, and connects them to different mechanical characteristics. Even though we have focused on the blood form of *T. brucei*, models for other trypanosome types [3] can quickly be established. Nevertheless, future simulation efforts have to be closely matched by the corresponding experimental research, in order to significantly advance our understanding of trypanosome-host interactions, which may also have clinical impact.

## Supporting information

**S1 Appendix. Measurement of the bending stiffness of the flagellum in simulations.**
(PDF)

**S2 Appendix. Description of the smoothed dissipative particle dynamic method [28–30] used for fluid modeling.**
(PDF)

**S1 Video. Simulation of trypanosome locomotion with parameters given in Table 1 for** $L_{flag}/\lambda_{in} = 2$ **and** $a_b/s_0 = 0.11$**.** The parasite body surface is represented by blue particles, while the flagellum by orange and green particles. The flagellum is embedded into the body and

its path includes a half turn around the body with a free part at the anterior end. The orange particles represent the two active filaments that define the beating plane. Swimming motion toward the anterior end also induces rotation of the parasite around its swimming axis.
(MP4)

**S2 Video. Persistent motion of a bloodstream form of *T. brucei* in fluid with a viscosity of** $5\ mPa \cdot s$**.** The video is recorded with a frame rate of 100 fps.
(MP4)

**S3 Video. Simulation of trypanosome propulsion with an increased flagellum rigidity of** $K = 2.43\ nN\mu m^2$**.** While the swimming velocity is only slightly increased in comparison to the case in S1V, the parasite rotation frequency is significantly enhanced.
(MP4)

**S4 Video. Simulation of parasite motion with a decreased flagellum stiffness of** $K = 0.243\ nN\mu m^2$**.** In this case, both the swimming velocity and rotation frequency are drastically reduced in comparison to the case in S1V.
(MP4)

**S5 Video. Trypanosome locomotion with a decreased body stiffness of** $\mu_b L_{tryp}^2 / k_B T = 7.8 \times 10^6$**, which leads to an increase in both the swimming velocity and rotation frequency when compared to the reference case in S1V.**
(MP4)

**S6 Video. Parasite propulsion with an increased body rigidity of** $\mu_b L_{tryp}^2 / k_B T = 7.8 \times 10^7$**.** While the rotation frequency is strongly reduced in comparison to the reference case in S1V, the swimming velocity is only slightly decreased.
(MP4)

**S7 Video. Simulation of parasite motion with an increasing wave length for flagellum actuation toward the anterior end.** In this case, wave amplitude at the anterior end becomes larger, leading to an increased swimming speed.
(MP4)

**S8 Video. Simulation of trypanosome propulsion with a decreasing wave length for flagellum actuation toward the anterior part.** Wave amplitude at the anterior end is smaller than in the reference case of S1V, such that the swimming velocity decreases.
(MP4)

**S9 Video. Trypanosome model with a flagellum beating plane normal to the body surface.** Here, the green particles represent the two active filaments.
(MP4)

## Acknowledgements

F.A.O., G.G., and D.A.F. gratefully acknowledge computing time on the supercomputer JURECA [50] at Forschungszentrum Jülich under grant no. actsys.

## Author contributions

**Conceptualization:** Gerhard Gompper, Dmitry A Fedosov.

**Formal analysis:** Florian A. Overberg, Narges Jamshidi Khameneh.

**Funding acquisition:** Markus Engstler, Gerhard Gompper, Dmitry A Fedosov.

**Investigation:** Florian A. Overberg, Narges Jamshidi Khameneh, Timothy Krüger.

**Methodology:** Florian A. Overberg.

**Project administration:** Markus Engstler, Dmitry A Fedosov.

**Software:** Florian A. Overberg, Dmitry A Fedosov.

**Supervision:** Timothy Krüger, Markus Engstler, Dmitry A Fedosov.

**Visualization:** Florian A. Overberg, Narges Jamshidi Khameneh.

**Writing – original draft:** Florian A. Overberg.

**Writing – review & editing:** Narges Jamshidi Khameneh, Timothy Krüger, Markus Engstler, Gerhard Gompper, Dmitry A Fedosov.

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
