## [Decision Letter · Decision Letter 0]

19 Feb 2025

PCOMPBIOL-D-24-02058

Modelling Trypanosome Motility

PLOS Computational Biology

Dear Dr. Fedosov,

Thank you for submitting your manuscript to PLOS Computational Biology. After careful consideration, we feel that it has merit but does not fully meet PLOS Computational Biology's publication criteria as it currently stands. Therefore, we invite you to submit a revised version of the manuscript that addresses the points raised during the review process.

Please submit your revised manuscript within 60 days Apr 21 2025 11:59PM. If you will need more time than this to complete your revisions, please reply to this message or contact the journal office at ploscompbiol@plos.org. Please include the following items when submitting your revised manuscript:

We look forward to receiving your revised manuscript.

Kind regards,

Anders Wallqvist

Academic Editor

PLOS Computational Biology

Amber Smith

Section Editor

PLOS Computational Biology

**Journal Requirements:**

At this stage, the following Authors/Authors require contributions: Florian Overberg, Narges Khameneh, Timothy Krüger, Markus Engstler, Gerhard Gompper, and Dmitry A Fedosov. Please ensure that the full contributions of each author are acknowledged in the "Add/Edit/Remove Authors" section of our submission form.

4) We notice that your supplementary information (Appendix A and Appendix B) is included in the manuscript file. Please remove them and upload them with the file type 'Supporting Information'. Please ensure that each Supporting Information file has a legend listed in the manuscript after the references list.

**Reviewers' comments:**

Reviewer's Responses to Questions

Reviewer #1: Some suggestions and recommendations:

1. The title "Modelling Trypanosome Motility" suggests a generic model, however, the simulations are based on the analysis of T. brucei motility. It is important to specify how the model could be adapted for different Trypanosoma species, as morphological variations among species may influence motility.

2. The authors state that the developed model realistically mimics the swimming behavior of trypanosomes; however, the model was created without considering the presence of cells in the medium. It is known that certain species, e.g., T. cruzi trypomastigotes, can detect the presence of mammalian cells and alter their motility patterns. Therefore, it would be valuable for the authors to include a discussion on the limitations of their model and clarify that it is initially applicable to scenarios where the parasite swims freely in the fluid. Additionally, they should discuss how interactions with the cellular environment affect motility and the model's parameters.

3. The proposed modeling approach may be challenging due to the parameters involved. Using artificial intelligence (AI) techniques based on experimental data would allow for more robust parameter adjustments, optimizing the model and improving its adaptability to different species. One suggestion would be for the authors to dedicate a step-by-step section explaining how this could be achieved.

Reviewer #2: This paper describes a three dimensional simulation of a Trypanosoma brucei cell, used to simulate how flagellum beating leads to swimming behaviour.

It is similar but represents an improvement on the previous model two of the authors were previously involved with (Alizadehrad et al 2015) by being a higher resolution simulation, and avoiding some key failures to accurately capture the cell shape. Specifically, having the flagellum run on the outside of the approximately helical cell shape, and having a beat amplitude comparable to real swimming cells. But, it also has some of the same failings: Specifically, not trying to consider the likely highly anisotropic mechanical properties of the cell body cytoskeleton. Additionally, it anchors beat plane orientation to the cell body, and appears to have incorrectly parameterised the trypanosome width.

Overall, the paper is clearly written, from my more biological background appears methodologically sound, and draws reasonable conclusions from the simulations and data presented. However, I would like to highlight two major limitations for two of the major conclusions:

Regarding contribution of cell body deformability to cell movement: 1) Not analysing the anisotropic cell body cytoskeleton misses one of the very characteristic properties of trypanosome cell organisation.

Regarding the flagellum beat plane orientation relative to the cell body: 2) Analysing a flagellum beating with a fixed beating plane relative to the cell body likely does not reflect the biomechanical properties of the actual flagellum-cell body connection.

Major comments and queries

Methodology

The cell body is simulated as an elastic mesh. The overall geometry is broadly reasonable, however seems to give a very wide cell, despite Rmax being set to 1.5 um. See, for example, the side by side comparison in Figure 3. Based on the scale bar in Fig 3, the actual diameter is perhaps 6 um instead of the target 3 um, unless there are other scaling issues.

I imagine that scaling the cell body diameter to correct this would not fundamentally change the results, but would shift several plots - this needs to be carefully checked.

The isotropic triangulation and harmonic spring potential used to model the cell surface cytoskeleton doesn't capture the highly anisotropic organisation of the parasite cytoskeleton (eg. Sinclair and de Graffenried 2020). The cell surface is underlain by a parallel array of microtubules maintained at approximately constant spacing - microtubules are relatively stiff and incompressible, and the consistent spacing suggests a microtubule-microtubule distance constraint. This likely gives anisotropic mechanical property of the cell surface, which is not captured. I can find no mention of the sub-pellicular microtubules of the cytoskeleton, and no citation of corresponding literature. There are relevant recent pieces of work from the de Graffenried, He, Vaughan and Varga research groups, older work from the Gull group, and original electron microscopy prominently from Keith Vickerman.

At minimum, this needs to be better explained and discussed. And would it be possible to test an anisotropic situation? There is sub-pellicular microtubule arrangement data available for T. brucei procyclic forms from expansion microscopy (Gorilak et al 2021) and Leishmania promastigotes from electron tomography (Hair et al 2024) as guides.

The energy potential for control of surface area is surely redundant, given the triangulated springs of the cell surface? Is there a way to probe its contribution, which would add confidence that the modelling of the cell surface as isotropic springs is providing the expected constraint, rather than this additional surface area constraint?

A simple way to test this may be to simply remove the surface area constraint and see similar behaviour.

Figure 5/Body and flagellum deformability

My concerns about 1) cell width scaling and 2) failure to capture the anisotropic parasite cytoskeleton mechanical properties limit the accuracy of any conclusions drawn here.

Figure 7 and 8/Tangential versus normal beating plane of the flagellum.

It is stated that "The orientation of the flagellum beating with respect to the body surface is very difficult to extract experimentally", but this is not true. Perhaps, difficult to determine from high speed videos. But flagellum bending occurs perpendicular to the midline between the axoneme central pair, and the orientation of the central pair is trivial to observe from electron microscopy. Similarly, the attachment plaque of the flagellum attachment zone, and its position relative to the axoneme, is trivial to observe from electron microscopy.

The model uses a fixed beat orientation to the cell body, achieved by anchoring both of either the oscillating or non-oscillating filaments as part of the cell body mesh. In biological terms, this corresponds to assigning the anchoring of the flagellum to the cell body via the flagellum attachment zone to a fixed doublet of the axoneme, and thus fixing the orientation of attachment relative to the plane of flagellum bending. In practice, the FAZ position around the axoneme is variable - Wheeler et al 2013 Figure 3 conveniently has several examples, where the point of attachment varies between ~ doublet 6 and 8, corresponding to a ~90 degree rotation of the axoneme, and thus rotation of bending orientation, relative to the cell body. From this quick observation, it is evident that the axoneme is free to rotate such that the beat plane can be either perpendicular or normal to the attachment to the cell body.

This is a major limitation for the results illustrated in Figure 8. The result of the simulation is interesting, but does not reflect the reality that the axoneme actually appears able to rotate by 90 degrees relative to the FAZ.

It would be much more informative to have the axoneme able to freely rotate relative to the cell body, then ask what orientation of beat it then prefers to attain. This could be achieved by changing anchoring of the axoneme to the cell to instead use an axoneme midline of vertices, which are then attached to the four axoneme filaments by radial connections. At minimum, this key limitation needs to be acknowledged.

Minor points

The cell volume constraint is important. It is mentioned qualitatively that volume preservation vs. considering a viscoelastic cytoplasm is likely to have a small effect, and I agree that the difference likely is small, but a quantitative statement would be nice if possible.

"What are the available experimental observations" which define the path of attachment of the flagellum to the cell body. I am more familiar with the procyclic life cycle stage morphology, and the attachment path does not match my expectations from that life stage. This work is, of course, describing the bloodstream form instead - but this raises a minor concern about the accuracy of this aspect of the model, and it is not a tested aspect of the model.

Was the viscosity of methylcellulose-HMI9 measured? If so how, and if not, how was this viscosity inferred?

Please carefully make it clear what is experimental data vs. simulation, as a prominent example, Figure 2 legend presents simulation as "Swimming characteristics of a trypanosome".

"Since the trypanosome flagellum is accompanied by a paraflagellar rod running in parallel, we chose a K value that represents a relatively stiff flagellum." Perhaps true, but worth expanding to note that the PFR runs along doublets 4 to 6, it is stretched and squashed by beating (even sitting on the inside of the more tightly curved bend of the asymmetric Leishmania beat, Wang et al 2020). You should also consider the biomechanical spring model of PFR function (Hughes et al 2012) - while I think this is unlikely to be correct, it is worth explicit mention.

Relating to Figure 6, is there any biological basis for differences in the proximal flagellum which might confer a larger or smaller actuation force? Eg. differences in axoneme composition.

There are several places where "in agreement with experimental observations" is stated with no reference. I realise that these tend to be quite general statements, but some reference really is required, even if just a paper with a good illustrative video rather than specifically talking about that point.

I am coming from a biological background, and I acknowledge some bias from this, but there was limited citation of trypanosome cell biology research groups with interests in motility and associated cell morphology - particularly He, Wheeler and Hill - and a tendency for self-citation of work from the Engstler group.

It is stated that "the unactuated shape of the trypanosome flagellum is not known". This is true, but really is very likely to be straight due to the rigidity of the microtubule-based structure. It would also be easy to test experimentally by lysis of trypanosomes and treatment with Ca2+ to depolymerise the sub-pellicular microtubules.

Reviewer #3: This paper describes a numerical model of T. brucei, a eukaryotic parasite of humans, livestock and other animals. The authors build a simulated cell body using a mesh of particles and add a simplified model flagellum to the cell body, composed of four filaments, an opposing pair of which are actuated to produce a flagellar beat. The authors investigate the shape of the flagellar beat (output wavelength, beat plane) as a function of various external and internal quantities including bending stiffness of the flagellum, amplitude of the input waveform etc. They conclude by outlining some future directions of research.

This is a nicely executed technical study on an important organism. I'm not too familiar with the numerical details of the simulations, but I have seen the SDPD before and it seems to be a well-established method. The parameter space is large but the authors make reasonable choices about which regimes to investigate, to maintain biological relevance. The results look sound to me, and this will be a useful tool in future studies, but I have a few queries around the application of the method and its interpretation.

1. The model includes harmonic potentials between 'atoms' within the cell body and within the flagellum; this implies that there is elasticity in the cell body but I can't see that there is a viscosity. Presumably, given the highly heterogeneous composition of cell membrane and cytoplasm, the body will contribute a viscous as well as elastic response, and that this viscous response (as opposed to interactions with the external fluid) may dominate the dissipation. Can the authors comment on this? Viscosity might lead to consequences including a phase lag between actuation and response that significantly alters the beat plane, body shape etc. It may simply be that incorporating a viscoelastic response in the cell body would make the computational requirements a lot more challenging - in which case, there is little choice but to leave it out.

2. The paraflagellar rod is mentioned in the introduction but then omitted in the later discussions. This structure is located next to the axoneme in the real cell and presumably lends an anisotropic bending stiffness to the flagellum in the real-world case. In fact, it seems that the PFR varies in length throughout the life cycle, both in absolute terms and relative to the rest of the flagellum (e.g. Bastin, Mol. Cell. Biol. 19(12) p.8191, 1999). Could the authors comment in its potential relevance in the discussion/conclusion section?

3. In the discussion, the authors state that their simulations 'confirm the hypothesis that trypanosome propulsion is primarily generated by the anterior part of the flagellum'. The simulations are consistent with this hypothesis, but 'confirm' is too strong. The parameters were chosen to mimic real life, so claims of 'confirmation' appear to be circular logic. I would suggest softening the claim.

4. When observing Trypanosomes under the microscope, I've been surprised by cells that appear to move vigorously, but make little forward progress. This isn't something for the current manuscript, but I wonder if this numerical model can help form mechanical hypotheses for why the cells exhibit this behaviour, and whether it's some sort of adaptation for another environment with different mechanical properties.

Minor issue

1. Could the authors put a dashed line with a slope of SP^-4 on Figure 4, as a guide to the eye?

**Have the authors made all data and (if applicable) computational code underlying the findings in their manuscript fully available?**

Reviewer #1: Yes

Reviewer #2: Yes

Reviewer #3: Yes

PLOS authors have the option to publish the peer review history of their article (what does this mean?). If published, this will include your full peer review and any attached files.

Reviewer #1: No

Reviewer #2: No

Reviewer #3: No

**Figure resubmission:**
---

## [Decision Letter · Decision Letter 1]

18 Apr 2025

PCOMPBIOL-D-24-02058R1

Modelling Motility of Trypanosoma brucei

PLOS Computational Biology

Dear Dr. Fedosov,

Thank you for submitting your manuscript to PLOS Computational Biology. After careful consideration, we feel that it has merit but does not fully meet PLOS Computational Biology's publication criteria as it currently stands. Therefore, we invite you to submit a revised version of the manuscript that addresses the points raised during the review process.

Please submit your revised manuscript within 30 days Jun 18 2025 11:59PM. If you will need more time than this to complete your revisions, please reply to this message or contact the journal office at ploscompbiol@plos.org. Please include the following items when submitting your revised manuscript:

We look forward to receiving your revised manuscript.

Kind regards,

Anders Wallqvist

Academic Editor

PLOS Computational Biology

Amber Smith

Section Editor

PLOS Computational Biology

**Reviewers' comments:**

Reviewer's Responses to Questions

**Comments to the Authors:**

Reviewer #2: Thank you for your comprehensive efforts to address my comments. I find the text markedly improved, particularly regarding discussions of some limitations/simplifications, referencing of the wider biological literature to place comments in better context, and the clarify of explanation of some methodological areas. Arguments against further simulation to test, for example, PFR stiffness or cortical cytoskeleton anisotropy are reasonable given lack of experimental data for the corresponding physical properties - although perhaps it would be nice for simulation to lead experiments here.

However, my key concern of cell size has not been addressed. The authors assure that the size of the model trypanosome matches reality, and have adjusted scale of some figure panels. However this does not address the root concern here.

I measured the width/length ratio from the model and microscopy images in Figure 3. I get Rmax ~= 0.08 * Ltryp for the model, larger than the stated 0.0625 from Table 1 presumably due to some out-of-plane bending. I get Rmax ~= 0.04 * Ltryp for the microscopy images.

A similar near two-fold mismatch in cell width can be seen by just overlaying the model and microscopy images, having scaled to roughly match length. I don't seem to be able to attach an image to illustrate this, so please do try the overlay and observe.

Impact of cell width on the simulation is not tested, so critical to observe and set it correctly. The width of the cell stated in Table 1 is Rmax = 1.5um. Why was this value selected? What references or measurements have guided this?

I'm afraid that statements like "the modelled body shape closely captures geometrical features of real T. brucei cells" cannot be true when simple observation suggests a two-fold difference in the width/length ratio.

Reviewer #3: I thank the authors for their careful consideration of my comments. In particular, the responses regarding the presence of a viscous fluid throughout the simulation volume, and those about the PFR, help to clarify the setup of their numerical work. I look forward to hearing more about how such numerical models might help in unpicking the biological context of surprising aspects of parasite swimming (e.g. the forward/backward wave 'cancelling'), and have no further comments or concerns regarding this manuscript.

**Have the authors made all data and (if applicable) computational code underlying the findings in their manuscript fully available?**

Reviewer #2: Yes

Reviewer #3: Yes

PLOS authors have the option to publish the peer review history of their article (what does this mean?). If published, this will include your full peer review and any attached files.

Reviewer #2: No

Reviewer #3: No

**Figure resubmission:**
---

## [Editor Report · Decision Letter 2]

2 May 2025

Dear Dr. Fedosov,

We are pleased to inform you that your manuscript 'Modelling Motility of Trypanosoma brucei' has been provisionally accepted for publication in PLOS Computational Biology.

Best regards,

Anders Wallqvist

Academic Editor

PLOS Computational Biology

Amber Smith

Section Editor

PLOS Computational Biology

---

## [Editor Report · Acceptance letter]

PCOMPBIOL-D-24-02058R2

Modelling Motility of Trypanosoma brucei

Dear Dr Fedosov,

I am pleased to inform you that your manuscript has been formally accepted for publication in PLOS Computational Biology. Your manuscript is now with our production department and you will be notified of the publication date in due course.

With kind regards,

Anita Estes
